# Metastatic Melanoma Progression Is Associated with Endothelial Nitric Oxide Synthase Uncoupling Induced by Loss of eNOS:BH4 Stoichiometry

**DOI:** 10.3390/ijms22179556

**Published:** 2021-09-03

**Authors:** Fabiana Henriques Machado de Melo, Diego Assis Gonçalves, Ricardo Xisto de Sousa, Marcelo Yudi Icimoto, Denise de Castro Fernandes, Francisco R. M. Laurindo, Miriam Galvonas Jasiulionis

**Affiliations:** 1Pharmacology Department, Universidade Federal de São Paulo, São Paulo 05508-090, Brazil; 2Department of Pharmacology, Institute of Biomedical Science, University of São Paulo, São Paulo 05508-060, Brazil; 3Micro-Imuno-Parasitology Department, Universidade Federal de São Paulo, São Paulo 05508-090, Brazil; diego.assis@icb.ufjf.br; 4Parasitology Department, Microbiology and Immunology, Federal University of Juiz de Fora, Juiz de Fora 36036-900, Brazil; 5Department of Physiological Sciences, Santa Casa de São Paulo School of Medical Sciences, São Paulo 01221-020, Brazil; ricardo.sousa@aluno.fcmsantacasasp.edu.br; 6Biophysics Department, Universidade Federal de São Paulo, São Paulo 05508-090, Brazil; icimoto@unifesp.br; 7Vascular Biology Laboratory, Heart Institute (InCor), University of São Paulo School of Medicine, São Paulo 05508-060, Brazil; denisecf11@gmail.com (D.d.C.F.); francisco.laurindo@hc.fm.usp.br (F.R.M.L.)

**Keywords:** metastatic melanoma, eNOS uncoupling, tetrahydrobiopterin, L-sepiapterin, eNOS:BH4 stoichiometry

## Abstract

Melanoma is the most aggressive type of skin cancer due to its high capability of developing metastasis and acquiring chemoresistance. Altered redox homeostasis induced by increased reactive oxygen species is associated with melanomagenesis through modulation of redox signaling pathways. Dysfunctional endothelial nitric oxide synthase (eNOS) produces superoxide anion (O_2_^−•^) and contributes to the establishment of a pro-oxidant environment in melanoma. Although decreased tetrahydrobiopterin (BH4) bioavailability is associated with eNOS uncoupling in endothelial and human melanoma cells, in the present work we show that eNOS uncoupling in metastatic melanoma cells expressing the genes from de novo biopterin synthesis pathway *Gch1*, *Pts,* and *Spr*, and high BH4 concentration and BH4:BH2 ratio. Western blot analysis showed increased expression of *Nos3*, altering the stoichiometry balance between eNOS and BH4, contributing to NOS uncoupling. Both treatment with L-sepiapterin and eNOS downregulation induced increased nitric oxide (NO) and decreased O_2_^•^ levels, triggering NOS coupling and reducing cell growth and resistance to *anoikis* and dacarbazine chemotherapy. Moreover, restoration of eNOS activity impaired tumor growth in vivo. Finally, *NOS3* expression was found to be increased in human metastatic melanoma samples compared with the primary site. eNOS dysfunction may be an important mechanism supporting metastatic melanoma growth and hence a potential target for therapy.

## 1. Introduction

Cutaneous melanoma originates from melanocytes, neural-crest-derived pigment-producing cells located in the epidermis. Even though the development of targeted therapy and immunotherapy has improved the overall survival of melanoma patients, its incidence continues to rise around the world, caused by increased life expectancy and ultraviolet radiation exposure [1,2,3,4]. Although melanoma occurrence is low compared to other types of cutaneous malignancies, it is responsible for approximately 75% of all skin-cancer-related deaths, which is associated with its chemoresistance to available therapeutic approaches and high ability to metastasize.

Different mechanisms associated with melanoma pathogenesis have already been described. Data from our group and others have shown that the establishment of a pro-oxidant intracellular milieu contributes to melanocyte malignant transformation through modulation of signaling pathways that favor proliferation and cell survival, and inhibit apoptosis, as was shown for other skin cancers [5,6,7,8,9]. The contribution of DNA and protein damage in the carcinogenesis induced by UV-derived ROS is also well established in non-melanoma skin cancers [10,11]. It was shown that the activation of RAS/RAC1/ERK signaling pathway in melanocytes submitted to a chronic stress condition is associated with oncogenesis, through regulation of global DNA methylation and DNMT1 expression. Moreover, melanocyte malignant transformation was severely impaired in the presence of a superoxide anion (O_2_^−•^) scavenger, showing the involvement of altered redox activity in melanomagenesis [7]. Dysfunctional mitochondria and increased activity of NADPH oxidase contribute to the disturbance of redox homeostasis in melanoma cells by raising reactive oxygen species (ROS) levels [12,13,14]. The involvement of nitric oxide synthases (NOSs) in redox landscape alteration has also been described in tumor cells [6,9].

NOSs are homodimeric enzymes that catalyze nitric oxide (NO) synthesis from L-arginine and molecular oxygen. The NOS reductase domain has homology with the cytochrome P450 enzyme, generating electrons flowing from the NADPH through the flavin domains FAD and FMN, and then transferred to the heme group of the oxidase domain of the other monomer, where L-arginine oxidation occurs. Integrated mechanisms, including activation by calcium–calmodulin, post-translational modifications as phosphorylation and glutathionylation, and substrate and cofactor availability, regulate NOS activity [15,16,17].

A critical aspect of NOS function is the bioavailability of the tetrahydrobiopterin cofactor BH4. BH4 is an unstable molecule that is easily oxidated to 7-8-dihydrobiopterin (BH2), which binds to the active enzyme site with the same affinity and is not able to catalyze NO formation. Consequently, reduction in BH4 concentration or BH4:BH2 ratio results in enzyme destabilization and dimer conformation disruption, inducing NOS dysfunction and reducing NO production. This state is referred to as uncoupling because NADPH oxidation and reduction of molecular oxygen are uncoupled from arginine hydroxylation and NO formation. However, the electron transfer from NADPH through the flavins domains to molecular oxygen is not inhibited, resulting in the generation of O_2_^−•^ [18,19]. Moreover, it was shown that in endothelial cells, not only the levels of BH4 and BH2, but also eNOS:BH4 stoichiometry determines eNOS function [20,21].

Increased expression of the three isoforms of *NOS* has been described in numerous tumor cells and has been related to carcinogenesis [9,22,23,24]. In melanoma, increased *NOS2* expression was correlated with worse prognosis and poor survival [25,26]. Other authors have demonstrated elevated expression of eNOS and nNOS [22,27] in melanoma tissues. However, these studies do not always address if the enzyme is functional and if it is producing NO or O_2_^−•^.

Previous data from our group suggested that eNOS uncoupling is a source of O_2_^−•^ in melanocytes submitted to anchorage blockade, since L-NAME, an eNOS inhibitor, and L-sepiapterin, a BH4 precursor, decreased O_2_^−•^ and increased NO levels [5,6]. The involvement of eNOS uncoupling in melanocyte malignant transformation was also shown, since L-NAME abrogated melanocyte transformation induced by sustained stress conditions. Besides that, inhibition of O_2_^−•^ production by L-sepiapterin treatment rendered metastatic melanoma cells less resistant to *anoikis* and more sensitive to carboplatin chemotherapy [6], showing the participation of NOS uncoupling in capabilities acquired during melanoma progression.

To investigate the contribution of uncoupling NOS and the mechanisms associated with its dysfunction in melanoma development, a murine melanoma model, comprising non-tumorigenic melan-a melanocytes, pre-malignant 4C melanocytes, non-metastatic 4C11− melanoma cells, and metastatic 4C11+ melanoma cells were used [28]. Based on the results here presented, we suggested that altered intracellular eNOS:BH4 stoichiometry is a key determinant of eNOS uncoupling in metastatic melanoma cells and is associated with melanoma aggressiveness.

## 2. Results

### 2.1. Increased Expression of de Novo Tetrahydrobiopterin Biosynthesis Enzymes during Melanoma Progression

Our group has previously showed decreased NO and increased O_2_^−•^ levels in cell lines corresponding to distinct stages of melanoma progression. In metastatic melanoma cells, L-sepiapterin treatment reversed this pattern, inducing NO production and suppressing O_2_^−•^, suggesting BH4 restoration and NOS coupling [6]. Intracellular BH4 amount is mainly controlled by the activity of the de novo pathway comprising the enzymes GTP cyclohydrolase 1 (GTPCH1), the rate-limiting enzyme in BH4 biosynthesis; 6-pyruvoyltetrahydrobiopterin synthase (PTPS); and sepiapterin reductase (SR) [19]. To elucidate the mechanisms underlying NOS uncoupling, the expression of the genes encoding these three enzymes was analyzed by real-time qPCR in melan-a melanocytes, 4C pre-malignant melanocytes, 4C11− non-metastatic melanoma cells, and 4C11+ metastatic melanoma cells. Surprisingly, the expression of *Gch1* was found to be increased in 4C11− and 4C11+ tumorigenic cells when compared with melan-a melanocytes and 4C pre-malignant melanocytes (Figure 1A), while *Pts* and *Spr* mRNA levels were augmented only in the metastatic 4C11+ cell line (Figure 1B,C). In endothelial cells and hepatocytes, it was demonstrated that *Gch1* activity, and consequently BH4 synthesis, is allosterically regulated by the enzyme GTP feedback regulator, GFRP. Interestingly, *Gchfr* expression was found increased in 4C pre-malignant melanocytes and 4C11− melanoma cells; however, it is reduced in 4C11+ metastatic melanoma cells when compared with melan-a melanocytes (Figure 1D). Data sets from TCGA showed increased *GCH1* expression along with melanoma progression (n = 248) (Appendix A) and decreased expression of *GCHFR* in melanoma (n = 461) compared with normal skin (n = 558) (Appendix A) [29]. In a study published by Talantov and colleagues, high *PTS* (Appendix A) and *SPR* levels (Appendix A) were observed in cutaneous melanoma (n = 45) compared with benign nevi (n = 18), corroborating the data found in our melanoma model [30] (GSE3189).

### 2.2. Tetrahydrobiopterin Concentration Is Elevated in Melanoma Cells

In endothelial cells, increased *Gch1* expression is associated with elevated BH4 levels [31,32]. Since increased expression of all the enzymes from the de novo synthesis pathway was observed in melanoma cells, the concentration of biopterins was analyzed by HPLC. BH4 and biopterins concentration was augmented along with melanoma progression, being higher in melanoma cells (Figure 2A,D), which corroborates the increased activity of the de novo BH4 synthesis cycle. However, BH2 amount did not change in all cells evaluated (Figure 2B). BH4:BH2 ratio was higher in 4C pre-malignant melanocytes and in 4C11− e 4C11+ melanoma cells when compared to melanocytes (Figure 2C).

### 2.3. Increased Expression of Endothelial Nitric Oxide Synthase in Metastatic Melanoma Cells

It has been described that eNOS uncoupling can be caused paradoxically by the overexpression of *NOS3* in endothelial cells [20]. Since decreased BH4 levels or BH4:BH2 ratio were not observed in melanoma cells and increased expression of *Nos3* has been observed in melan-a melanocytes submitted to sustained stress condition, which resulted in melanocyte malignant transformation [5] and gave rise to the cells used in this study, we evaluated eNOS expression in these melanoma cell lines by real-time qPCR. Increased expression of *Nos3* was found in 4C11+ metastatic melanoma cells (Figure 3), suggesting that eNOS uncoupling could be a result of loss of eNOS:BH4 stoichiometry.

### 2.4. Downregulation of eNOS Protein Restores Its Function

To investigate if the overexpression of *Nos3* in 4C11+ metastatic melanoma cells was associated with its dysfunction and the altered redox status observed in these cells, *Nos3* was knocked down in 4C11+ melanoma cells using an eNOS shRNA. A scramble control (TRC1) or eNOS shRNA was introduced in 4C11+ cells and two clones (#1 and #2) with decreased *Nos3* expression were established as evaluated by RT-qPCR (Figure 4A) and Western blot (Figure 4B). The results suggest that downregulation of eNOS protein restores its function, since we observed increased NO (Figure 4C) and decreased O_2_^−•^ levels (Figure 4D) after eNOS silencing.

### 2.5. Decreased Expression and Coupling of eNOS in the Presence of a Superoxide Scavenger

Elevated ROS concentration induced the expression of eNOS, as shown at both the transcriptional and translational levels in endothelial cells [33,34]. Previous data from our group showed that O_2_^−•^ depletion impaired melan-a melanocyte malignant transformation induced by sequential cycles of anchorage blockade, giving rise to cells—namely Mn3, Mn4, and Mn5—with a senescent-like phenotype and low proliferation rate [7]. To determine if oxidative stress during melanoma progression contributes to increased *Nos3* expression found in 4C11+ metastatic melanoma cells, its expression was evaluated in Mn3 and Mn5 cells. We found decreased *Nos3* expression in Mn3 and Mn5 cells compared to 4C11+ metastatic melanoma cells (Appendix A). Moreover, eNOS activity seems to be restored, since O_2_^−•^ is reduced (Appendix A) and NO amount elevated (Appendix A) in Mn3 and Mn5 in comparison with 4C11+ melanoma cells.

### 2.6. L-Sepiapterin Supplementation Decreases Superoxide Anion and Increases Nitric Oxide Levels

It was reported that administration of BH4 in mice overexpressing eNOS in vivo prevented eNOS uncoupling by restoring NO production [20,35]. To evaluate if increased intracellular BH4 can restore eNOS function in 4C11+ metastatic melanoma cells, they were treated or not with 40 µM L-sepiapterin for two hours. In the presence of L-sepiapterin, decreased O_2_^−•^ levels were observed, as demonstrated by flow cytometer (Figure 5A) and HPLC analyses (Figure 5B). In addition, NO was increased after L-sepiapterin treatment as demonstrated by flow cytometer analyses (Figure 5C) and by NO analyzer (Figure 5D). These results corroborate with previous data from our laboratory, where a cell line isolated from a murine metastatic melanoma (Tm5 lineage) showed the same pattern [6]. L-sepiapterin treatment was efficient, since both increased intracellular BH4 and biopterin concentration (Appendix A) and BH4:BH2 ratio were observed (Appendix A).

### 2.7. Restoration of eNOS Function Reduced Cell Survival of Metastatic Melanoma Cells

Since reduced *Nos3* expression, as well as L-sepiapterin treatment in 4C11+ metastatic melanoma cells, induced restoration of its activity by increasing NO synthesis and decreasing O_2_^−•^, cell survival was evaluated. Downregulation of eNOS decreased cell viability of 4C11+ metastatic melanoma cells compared to wild type and scramble control cells (Figure 6A) for 72 and 96 h and colony formation in vitro (Figure 6B). eNOS silencing also conferred *anoikis* sensitivity to 4C11+ metastatic melanoma cells when compared to scrambled control cells (Figure 6C).

### 2.8. Downregulation of eNOS Attenuated Metastatic Melanoma Growth In Vivo

Since we observed decreased cell viability, colony formation, and *anoikis* resistance in 4C11+ metastatic melanoma cells silenced for eNOS, in vivo tumor growth was analyzed. Metastatic melanoma 4C11+ cells-wild type, scramble, and silenced for eNOS were inoculated in the subcutaneous flank of female C57BL/6 mice, and tumor development was observed. Tumors of all groups appeared after 15 days; however, the tumor development of 4C11+ metastatic melanoma cells silenced for eNOS was mitigated, as observed after 21 days when the animals were sacrificed and the tumor mass measured. Decreased eNOS expression caused a significant reduction in both tumor volume and weight (Figure 7A,B).

### 2.9. L-Sepiapterin Supplementation Inhibits the Growth of Metastatic Melanoma Cells

A pro-oxidant microenvironment sustained by eNOS uncoupling seems to contribute to the survival of metastatic melanoma cells, since L-sepiapterin treatment reduced cell viability at 24, 48, 72, and 96 h as shown by MTT (Figure 8A) and clonogenic assay (Figure 8B). Decreased O_2_^−•^ levels induced by L-sepiapterin supplementation also rendered 4C11+ metastatic melanoma cells less resistant to *anoikis* (Figure 8C). Moreover, L-sepiapterin increased 4C11+ melanoma sensitivity to dacarbazine chemotherapy, as shown in Figure 8D.

### 2.10. Increased NOS3 Expression in Human Metastatic Compared with Primary Melanomas

Analysis of Riker gene array with 87 samples from Oncomine showed high *NOS3* expression in metastatic melanoma when compared to the primary site (Figure 9A) [36] (GSE7553). Moreover, analysis from TCGA showed increased *NOS3* expression along with melanoma progression (Figure 9B) [29].

## 3. Discussion

Previous data from our group showed the establishment of a murine melanoma model where different melanoma cell lines were obtained after submitting melan-a melanocytes to serial cycles of anchorage blockade, which was associated with oxidative stress [28]. The involvement of O_2_^−•^ in the regulation of signaling pathways and epigenetic mechanisms associated with melanocyte malignant transformation was demonstrated [5,7]. Moreover, the participation of eNOS uncoupling was suggested since L-NAME and L-sepiapterin decreased O_2_^−•^ and increased NO during adhesion impediment. More importantly, the treatment of melan-a with L-NAME during de-adhesion impaired the acquisition of a malignant phenotype, reinforcing the role of uncoupled eNOS in melanoma development [6].

NOS function is regulated by several integrated mechanisms, with intracellular availability of the BH4 cofactor being essential [18]. Intracellular BH4 concentration is tightly controlled in different ways and is maintained by de novo, salvage, and recycle synthesis pathways (Figure 1E) [37]. In de novo BH4 biosynthesis, BH4 is synthesized from GTP in a three-step reaction as follows: GTPCH1 (EC 3.5.4.16), encoded by *GCH1* gene, is followed by 6-pyruvoyl-tetrahydrobiopterin synthase PTPS (EC 4.6.1.10) and sepiapterin reductase SR (EC 1.1.1.153). In endothelial cells and cardiomyocytes, it was shown that GTPCH1 activity can be regulated transcriptionally since *GCH1* mRNA expression is correlated with protein content and BH4 production [32,38,39]. Surprisingly, we found increased expression of *Gch1* mRNA in 4C11− and 4C11+ melanoma cells when compared with melan-a melanocytes and 4C pre-malignant melanoma cells (Figure 1A). Since we have previously suggested that eNOS uncoupling is associated with melanoma cell growth and apoptosis resistance [6], we expected to observe decreased *Gch1* expression and activity. However, this result corroborates the literature data, since *GCH1* expression is increased in estrogen-receptor-negative breast tumors and cells present in the tumor microenvironment, like stromal fibroblasts and inflammatory and endothelial cells, being a significant predictor of poor prognosis in patients [40]. Overexpression of GTPCH1 in glioblastoma is also associated with tumor growth and higher glioma grade, recurrence, and worse survival [41]. Epidermal melanocytes already express GTPCH1 in vitro and in vivo and can synthesize BH4, which is involved in melanogenesis [42]. Furthermore, *Gch1* mRNA expression is increased in the presence of high ROS concentration [43]. Therefore, these circumstances could explain the *Gch1* upregulation observed in melanoma cells. *Pts* and *Spr* mRNA expression were increased only in 4C11+ metastatic melanoma cells (Figure 1B,C). Expression of PTPS is augmented in the early phases of colorectal cancer and under hypoxia, and its induction in colon carcinoma cell lines was associated with tumor growth [44]. Moreover, PTPS downregulation decreased BH4 synthesis, demonstrating that altered expression of other enzymes from the BH4 synthesis pathway may also affect its production. In our melanoma model, we observed up-regulation of genes associated with hypoxia response along with melanoma progression [45]. Therefore, increased *Hif-1* expression in 4C11+ melanoma cells can be associated with *Pts* up-regulation. The overexpression of *SPR* mRNA was also found in neuroblastoma, breast cancer, and hepatocarcinoma cells [46,47,48]. It was demonstrated that SR is associated with tumor growth through different mechanisms, including interaction with ornithine decarboxylase, induction of FoxO3a/Bim signaling pathway, and oxidative stress. The mechanisms underlying the role of SR in melanoma development are under investigation. The de novo BH4 biosynthesis is tightly regulated by GFRP. By an allosteric mechanism, high BH4 concentration decreases GTPCH1 activity via interaction with GFRP, while L-phenylalanine stimulates BH4 production [49]. We observed increased *Gchfr* expression in 4C pre-malignant melanoma cells and 4C11− melanoma cells; however, in 4C11+ metastatic melanoma cells, *Gchfr* expression is reduced (Figure 1D). Under oxidative stress conditions, *Gchfr* mRNA is also upregulated in endothelial cells and cardiomyocytes [43]. Since O_2_^−•^ is increased along with melanoma progression [6], the overexpression of *Gchfr* in 4C pre-malignant melanocytes and 4C11− non-metastatic melanoma cells could be associated with altered redox homeostasis. However, decreased *Gchfr* expression observed in 4C11+ melanoma cells need more investigation. The functional GFRP/GTPCH1 axis has been described in epidermal melanocytes [42], suggesting that this feedback mechanism could be controlling BH4 synthesis in melanoma cells. However, it was shown in other systems that modifications in GFRP expression do not necessarily occur concomitantly with changes in GTPCH1 activity or BH4 synthesis [32]. Moreover, public data from Oncomine and TCGA showed that high *GCH1* expression correlates with melanoma aggressiveness (Appendix A), increased expression of *PTS* (Appendix A) and *SPR* (Appendix A), and decreased *GCHFR* expression (Appendix A) in melanoma when compared to benign nevi. According to the increased expression of the *Gch1* gene, BH4 and total biopterins concentration were increased in 4C11− and 4C11+ melanoma cells when compared to melanocytes (Figure 2A,D), showing that in the tumor, as demonstrated in other cells, GTPCH1 activity and BH4 production are regulated at the transcriptional level. BH4 is easily oxidized in the presence of high levels of ROS; however, we did not observe changes in BH2 concentration throughout the development of melanoma (Figure 2B), suggesting that the salvage and recycle pathways are functional, since these cells produce high levels of ROS [6]. Consequently, BH4:BH2 ratio was increased in 4C pre-malignant cells and 4C11− and 4C11+ melanoma cells (Figure 2C). Although this information is not always related to cancer cells, it would be relevant to evaluate BH4:BH2 ratio, since the accumulation of BH2 could induce eNOS uncoupling by competing with BH4 for eNOS active site. As mentioned earlier, although BH2 binds to the active site of eNOS with the same affinity, it does not have the ability to catalyze the formation of NO. Therefore, BH4:BH2 ratio in addition to the BH4 absolute molar concentration are key determinants of eNOS function [21,50,51]. Based on these results, it would be reasonable to think that NOS is not uncoupled in these cells. However, we have described a reduction in NO and an increase in O_2_^−•^ in Tm5 metastatic melanoma cells, which is in apparent contradiction with the results shown here. Moreover, L-sepiapterin treatment induced a switch in the redox status of melanoma cells, impairing O_2_^−•^ production and stimulating NO synthesis [6]. It was reported in vitro and in vivo that the stoichiometric relationship between eNOS and BH4 also determines eNOS activity. These authors demonstrated that *Nos3* superexpression in the absence of GTPCH1 activity induces uncoupling, increasing O_2_^−•^ levels and reducing NO synthesis [20,21]. The mechanism underlying uncoupling is that the amount of BH4 is not enough to bind overexpressed eNOS. In human melanoma cells, the results also suggested that the imbalance between NO and O_2_^−•^ is caused by the alteration in BH4:eNOS stoichiometry and decreased BH4:BH2 ratio [9]. We demonstrated increased *Nos3* mRNA expression in 4C11+ metastatic melanoma cells (Figure 3), as observed during melanocyte anchorage blockade impediment [5]. Therefore, increased ROS levels observed in 4C11+ melanoma cells could be a result of a disturbance between BH4 amount and *Nos3* expression. To evaluate this hypothesis, *Nos3* mRNA expression was downregulated in 4C11+ melanoma cells and two clones were settled down, sheNOS#1 and sheNOS#2 (Figure 4A,B). Decreased *Nos3* expression restored eNOS function by impairing O_2_^−•^ production (Figure 4C) and elevating NO levels (Figure 4D), as shown previously in endothelial cells [20]. Upregulation of *Nos3* expression has been associated with *K-Ras*-driven tumors, as pancreatic ductal adenocarcinoma and papillomas, since NO, through wild-type Ras S-nitrosylation, activates this signaling pathway, leading to tumor growth [52,53]. The participation of mutated *RAS* in melanoma development is well known [54,55]. There is no evidence that *N-RAS* mutations are caused by UV-induced DNA damage; however, RAS activation can be induced by increased ROS production, which is also a result of UV exposure [7,56]. In fact, we and others have shown that Ras is redox-sensitive, being associated with melanocyte malignant transformation and melanoma development. In the presence of MnTBAP, an O_2_^−•^ scavenger, Ras activity is decreased [7]. Therefore, increased *Nos3* expression may lead to uncoupling, altering cellular redox status, and overactivating the Ras transduction pathway, which, in turn, would contribute to melanomagenesis. Moreover, in B16F10 melanoma tumors from chronically stressed animals, *Nos3* is overexpressed and contributes to tumor development [57]. It was shown that eNOS expression is increased in the presence of high ROS levels in endothelial cells [33,34]. Therefore, increased *Nos3* expression could be a result of altered redox homeostasis found in 4C11+ metastatic melanoma cells. Corroborating this idea, we observed decreased *Nos3* expression (Appendix A), reduced O_2_^−•^ (Appendix A), and increased NO amount (Appendix A) in clones obtained after submitting melan-a cells to sequential cycles of anchorage impediment in the presence of MnTBAP, a superoxide scavenger [7]. MnTBAP treatment impaired an efficient malignant transformation, and one of the mechanisms underlying this phenomenon could be eNOS coupling. The restoration of NOS activity in 4C11+ melanoma cells was also observed after L-sepiapterin treatment, as indicated by decreased O_2_^−•^ (Figure 5A,B) and increased NO (Figure 5C,D), corroborating the hypothesis of eNOS uncoupling. Recovering redox homeostasis with L-sepiapterin treatment was already showed in vitro and in vivo, including in cancer models [6,58,59,60]. L-sepiapterin supplementation impaired chemically-induced murine colitis and colon cancer [59] and reduced breast tumor growth in vitro and in vivo [60] through the restoration of coupled NOS activity. Furthermore, BH4 and L-sepiapterin decreased human metastatic melanoma cell survival by inducing apoptosis [9]. In agreement with these studies, L-sepiapterin decreased 4C11+ cell viability (Figure 8A) and clonogenicity capability (Figure 8B). Furthermore, it also rendered 4C11+ melanoma cells more sensitive to *anoikis* (Figure 8C) and dacarbazine treatment (Figure 8D). Downregulation of *Nos3* also decreased cell viability (Figure 6A), clonogenicity capability (Figure 6B), and *anoikis* resistance of 4C11+ melanoma cells (Figure 6C). More important, reduced expression of eNOS in 4C11+ melanoma cells impaired tumor growth in vivo, as demonstrated by decreased tumor weight (Figure 7A) and volume (Figure 7B). Corroborating the role of eNOS in *RAS*-driven cancers, genetic ablation of *NOS3* gene disrupted pancreatic carcinoma and papilloma development and improved mice median survival. These effects were also shown after oral administration of L-NAME [53]. Finally, Oncomine and TCGA analysis showed that *NOS3* expression correlated with human melanoma progression (Figure 9), reinforcing the participation of eNOS in melanoma development. As far as we know, this is the first time that increased eNOS expression has been related to metastatic melanoma. It has been reported that eNOS expression is increased in melanoma tissue when compared with melanocytic nevi; however, the melanoma grade was not discriminated [22]. It is important to note that *Nos3* overexpression does not correlate with increased NO production, but instead O_2_^−•^ generation, which means that is fundamental to assess enzyme function to understand the real participation of any protein in pathological conditions.

## 4. Materials and Methods

### 4.1. Cell Culture

A murine melanocyte malignant transformation model was developed in our laboratory after submitting an immortalized but non-tumorigenic melan-a melanocytes to a sustained stress condition, giving rise to pre-malignant 4C melanocytes, non-metastatic 4C11− melanoma cells, and metastatic 4C11+ melanoma cells [7,28]. Melan-a cells were cultured at 37 °C in humidified 95% air/5% CO2 in RPMI pH 6.9 supplemented with 5% fetal bovine serum (Invitrogen, Scotland, UK), 200 nM 12-phorbol-13-myristate acetate (PMA; Calbiochem, Darmstadt, Germany), 100 U/mL penicillin, and 100 U/mL streptomycin (Invitrogen, Grand Island, NY, USA). Pre-malignant 4C melanocyte lineage and non-metastatic 4C11− and metastatic 4C11+ melanoma cell lines were cultured as melan-a cells, but in the absence of PMA. 4C11+ melanoma cells were treated or not with 40 μM L-sepiapterin (Cayman, Ann Arbor, MI, USA) for 2 hours for NO and O_2_^−•^ quantification; 96 h for BH4 quantification, MTT, and *anoikis* resistance assay; and 9 days for colony formation assay.

### 4.2. Stable Silencing of eNOS by shRNA

eNOS was silenced by shRNA using MISSION^®^ Lentiviral Transduction Particles from (Sigma-Aldrich, Saint Louis, MO, USA). Two shRNA sequences for eNOS (sheNOS#1, Clone ID: NM_008713.2-3844s1c1, and sheNOS1#2, Clone ID: NM_008713.2657s1c1) and a control plasmid containing a non-target sequence (non-Mammalian shRNA Control, TCR1, SHC002) were used. Lentiviral particles were produced from TRC1 and TRC1.5 libraries consisting of sequence-verified cloned into the pLKO-1-puro vector and were used to establish cell lines with stable eNOS knockdown or expressing the non-target shRNA sequence as a control. Metastatic melanoma 4C11+ cells were plated on a 24-well plate and maintained in complete medium for 24 h. Viral particles at MOI 1.0 (multiplicity of infection) were added to the cells, gently mixed, and incubated overnight. After this period, medium containing viral particles was replaced by a complete fresh medium for 24 h. In order to establish stable-silenced eNOS clones, puromycin-resistant cells were selected and submitted to clonal dilution.

### 4.3. Nitric Oxide Quantification

NO concentration was evaluated after a gas-phase chemiluminescence reaction of NO with ozone by an NO analyzer (NOA 280; Sievers Instruments, Boulder, CO, USA). A standard curve was established with a set of serial dilutions (0.1–100 μM) of sodium nitrate. NO metabolite concentrations in samples cells were settled by comparison with a standard curve and expressed as micromoles per milligram of protein using the NOANalysis software (version 3.21; Sievers Instruments, Inc., Boulder, CO). NO levels were also measured using a non-fluorescent indicator for nitric oxide, 4-5-diaminofluorescein diacetate (DAF-2DA, Molecular Probes, Eugene, OR, USA). Cells were incubated with 5 μM DAF-2DA at 37 °C for 30 min in the dark, rinsed with phosphate-buffered saline (PBS), and NO levels analyzed by flow cytometry (FACSCalibur; Becton-Dickinson, Franklin Lakes, NJ, USA) (excitation wavelength 495 nm; emission wavelength 515 nm).

### 4.4. Superoxide Anion Quantification

The amount of superoxide anion was measured using dihydroethidium (DHE; Molecular Probes, Eugene, OR, USA), a non-fluorescent cell-permeable indicator for O_2_^−•^. For flow cytometer analysis, cells were washed and incubated in PBS for 30 min at 37 °C, incubated with 25 μM DHE for 40 min at 37 °C in the dark, washed, and analyzed by flow cytometry (FACSCalibur; Becton–Dickinson, Franklin Lakes, NJ, USA) (excitation wavelength 480 nm; emission wavelength 567 nm). For high-performance liquid chromatography (HPLC) evaluation, cells were washed three times with PBS and incubated in PBS containing 100 μM DTPA (diethylenetriamine pentaacetic acid) (Sigma, St. Louis, MO, USA) and 50 μM DHE for 30 min at 37 °C. After washing, acetonitrile extraction was performed and simultaneous fluorescent detection of 2-hydroxyethidium (2-E+OH) and ethidium was done. DHE-derived product 2-E+OH was expressed as ratios of generated 2-E+OH over consumed DHE (initial DHE concentration minus remaining DHE) [61].

### 4.5. Biopterin Quantification

The concentrations of tetrahydrobiopterin (BH4), 7,8-dihydrobipterin (BH2), total biopterin, and BH4:BH2 ratio were analyzed by reversed-phase high-performance liquid chromatography as previously described [62]. Melan-a, 4C, 4C11− and 4C11+, and 4C11+ cells treated or not with 40 μM L-sepiapterin for 96 h were washed twice with cold PBS (pH 7.4). After centrifugation, cells were resuspended in 0.5 mL 0.1 M phosphoric acid containing 5 mM dithioerythritol (DTT) and sonicated for 40 s, and 35 µL 2 M trichloroacetic acid (TCA) was added. The solution was centrifuged at 12,000× *g* for 1 min, and the supernatant was used immediately for the quantification of all biopterins. The total biopterin amount was measured following oxidation in acidic conditions, whereas BH2 quantification was conducted after its oxidation in alkaline conditions. BH4 was calculated from the difference between the amount of biopterin formed by oxidation in acidic conditions and the amount of biopterin formed by oxidation in alkaline conditions. For oxidation reaction under acidic conditions, 100 µL cell extract was mixed with 15 µL 0.2 M TCA and 15 µL 1% I2/2% KI in 0.2 M TCA. For oxidation under alkaline conditions, 100 µL cell extract were mixed with 15 µL 1 M NaOH and 15 µL 1% I2/2% KI in 3 M NaOH. The oxidation reaction was carried out for 1 h in the dark at room temperature. To inactivate the excess of iodine, 25 µL 0.114 M ascorbic acid was added and centrifuged at 4 °C for 12 min, and 100 µL of the supernatant was injected into an HPLC system (LCMS-2020, Shimadzu Co., Kyoto, Japan) on a C18 Vydac reversed-phase column (5 μm, 4.6 mm id × 205) and detected by fluorescence (lex = 350 nm; lem = 450 nm). Biopterins were eluted by an isocratic mobile phase solution (5% methanol and 7.5 mM sodium phosphate buffer, pH 6.35) at a flow rate of 1.0 mL/min. Data were collected and analyzed by LC solution software (Shimadzu Co., Kyoto, Japan) and normalized against protein concentration.

### 4.6. RNAm Expression Analysis

RNA was isolated using Trizol (Invitrogen, Carlsbad, CA, USA) according to the manufacturer’s specifications. Equal amounts of each cDNA synthetized were quantified by real time-PCR in a Corbett Rotor-Gene 6000 Detection System version 1.7 using an SYBR green PCR master mix (Qiagen, Dusseldorf, German) and specific primers. Relative quantification (RQ) of the amplicons was calculated according to the 2-∆∆Cq method. Gene normalizing was performed using β-actin.

### 4.7. Western Blot

Subconfluent cell cultures were trypsinized and washed in PBS, and whole-cell lysates were prepared using NP-40 lysis buffer (10% NP-40 in 100 mM NaCl and 50 mM Tris pH 7.4, containing 30 mM sodium pyrophosphate, 50 mM NaF, 1 mM NaVO_4_, 2 mg/mL aprotinin, 2 mg/mL leupeptin, 2 mg/mL pepstatin, and 1 mM PMSF), kept for 15 min on ice, followed by centrifugation at 13,000× *g* for 15 min at 4 °C. The supernatant was collected for protein concentration using a Bio-Rad protein assay dye reagent (Bio-Rad, Hercules, CA, USA). Equivalent amounts of protein (40 μg) were denatured in SDS sample buffer (240 mM Tris–HCl pH 6.8, 0.8% SDS, 200 mM beta-mercaptoethanol, 40% glycerol, and 0.02% bromophenol blue) for 5 min. Protein lysates were resolved by SDS-PAGE and transferred onto nitrocellulose membranes (Bio-Rad, Hercules, CA, USA). The specific antibody used was a rabbit anti-eNOS (Cell Signaling Technology Danver, MA, USA) followed by incubation with an anti-rabbit conjugated to peroxidase (Bio-Rad, Hercules, CA, USA). The signal was visualized by chemoluminescence using the Immobilon Forte Western HRP substrate (Merck KGaA, Darmstadt, Germany). The band intensities were quantified using Processing and Analysis in Java, ImageJ 1.38b (Wayne Rasband, National Institute of Health, USA, http://www.rsb.info.nih.gov/ij/, accessed on 27 June 2021).

### 4.8. MTT Assay

The viability of 4C11+ melanoma cells—wild type, non-target (TRC1), sheNOS#1, sheNOS#2, and 4C11+ cells, treated or not with 40 μM L-sepiapterin—was evaluated using the standard MTT assay (Calbiochem, Hesse, Germany). Cells were harvest with trypsin and cultured on 96-well plates at 37 °C in a humidified atmosphere containing 5% CO_2_. Cell viability was analyzed every 96 h when 5 mg/mL MTT was added to the culture. Cells were kept in an incubator at 37 °C and 5% CO_2_ with MTT for one hour. After the medium was withdrawn, 100 μL of isopropanol (Merck, Hesse, Germany) was added to all wells and incubated for 15 min. Absorbance was quantified on a spectrophotometer at 620 nm (Multiskan EX, Labsystem, Vantaa, Finland). Cell viability was also evaluated in cells treated or not with 500 μM dacarbazine and 500 μM dacarbazine plus 40 μM L-sepiapterin for 48 h.

### 4.9. Colony Formation Assay

4C11+ melanoma cells—wild type, non-target (TRC1), sheNOS#1, sheNOS#2, and 4C11+ cells, treated or not with 40 μM L-sepiapterin—were harvested with mild trypsin treatment, and two hundred cells were plated on 60 mm dishes. After nine days, the plates were washed in PBS, fixed in 3.7% (*v*/*v*) formaldehyde for 15 min, washed with PBS, stained with 1% Toluidine Blue in 1% sodium tetraborate (borax) for five minutes, and washed with water. For the quantification of surviving cells, the dye was solubilized in 1% SDS under agitation for one hour, and the absorbance at 620 nm was measured using a spectrophotometer.

### 4.10. Anoikis Resistance Assay

To estimate *anoikis* resistance, 4C11+ melanoma cells—wild type, non-target (TRC1), sheNOS#1, sheNOS#2, and 4C11+, treated or not with 40 μM L-sepiapterin—were maintained for 96 h in 100 mm^2^ dishes coated with 1% agarose. After this time, cells were collected, centrifuged, and seeded in a 96-well plate. Cell viability was assessed by MTT after cell adhesion.

### 4.11. Tumor Growth Assay

Cells (2 × 10^5^) were subcutaneously injected into the flanks of 6-to-10-week-old C57BL/6 female mice (6 animals per group). Animals were kept under 12 h daylight cycles without water or food restriction, and tumor growth was observed and measured every two days. On day 21, mice were sacrificed, and the tumor weight was determined. Tumor volume (mm^3^) was measured using the formula d^2^ × D/2, where d represents the minimum diameter and D the maximum diameter. All procedures involving animals were performed after approval from the Research Ethics Committee of the Universidade Federal de São Paulo, Brazil (Approval no. 0219/07).

### 4.12. Statistical Analysis

Data analysis was performed for two groups by Student’s *t*-test, and for three or more groups by analysis of variance (factorial ANOVA) with Bonferroni post-test, using the Graphpad Prism 7.0^®^ statistical software (GraphPad, San Diego, CA, USA). The significance level was established at *p* < 0.05.

## 5. Conclusions

In conclusion, eNOS uncoupling is involved in the maintenance of a pro-oxidant milieu in melanoma cells, which in turn contributes to melanoma progression. Moreover, eNOS uncoupling in melanoma cells is a result of disrupted BH4:eNOS stoichiometry. Targeting eNOS in melanoma cells might be considered as a potential target therapy, and further studies in this field are warranted.

## Figures and Tables

**Figure 1 ijms-22-09556-f001:**
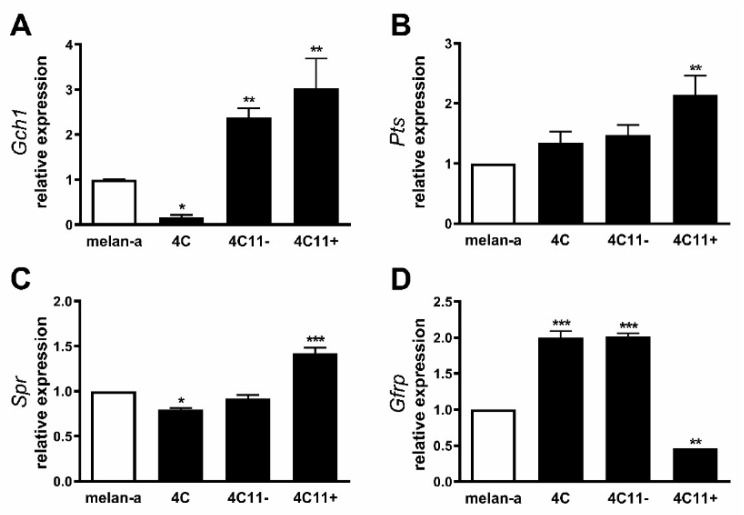
Expression of de novo tetrahydrobiopterin synthesis pathway along with melanoma progression. Relative mRNA levels of the enzymes from de novo tetrahydrobiopterin via (**A**) *Gch1*, (**B**) *Pts*, (**C**) *Spr*, and (**D**) *Gfrp* were determined in melan-a melanocytes (Ma), 4C pre-malignant melanocytes, 4C11− non-metastatic melanoma cells, and 4C11+ metastatic melanoma cells by real-time qPCR using specific primers. β-actin gene was used as an internal control. (**E**) Tetrahydrobiopterin biosynthesis pathways. In the de novo pathway, tetrahydrobiopterin (BH4) is synthesized by the sequential action of three enzymes from guanosine triphosphate (GTP). The first step is catalyzed by the reaction-limiting enzyme, GTP cyclohydrolase 1 (GTPCH1), which converts GTP into 7,8-dihydroneopterin triphosphate, which is converted by the action of the enzyme pyroviltetrahydrobiopterin synthase (PTPS) into 6-pyrovil tetrahydrobiopterin. The last reaction is catalyzed by the enzyme sepiapterin reductase (SR) which produces BH4 from 6-pyrovil tetrahydrobiopterin. BH4 can be spontaneously oxidized to dihydrobiopterin (BH2). To restore BH4, BH2 is reduced by the enzyme dihydrofolate reductase (DHFR) in the salvage pathway. BH4 can also be converted to quinonoid-BH2 (qBH2), an unstable molecule that is easily converted to BH2, being reduced to BH4 by the enzyme dihydropteridine reductase (DHPR). The SR enzyme also reduces exogenous sepiapterin to BH2, which is reduced by DHFR to BH4. GTP feedback regulator (GRFR) binds to GTPCH1 and regulates BH4 synthesis in a feedback mechanism. Values are reported in the bar graphs and expressed as the means ± S.D. The experiments were performed in triplicate and *p* values were based on the one-way ANOVA test followed by Bonferroni post-test; *, *p* < 0.05; **, *p* < 0.01; ***, *p* < 0.001.

**Figure 2 ijms-22-09556-f002:**
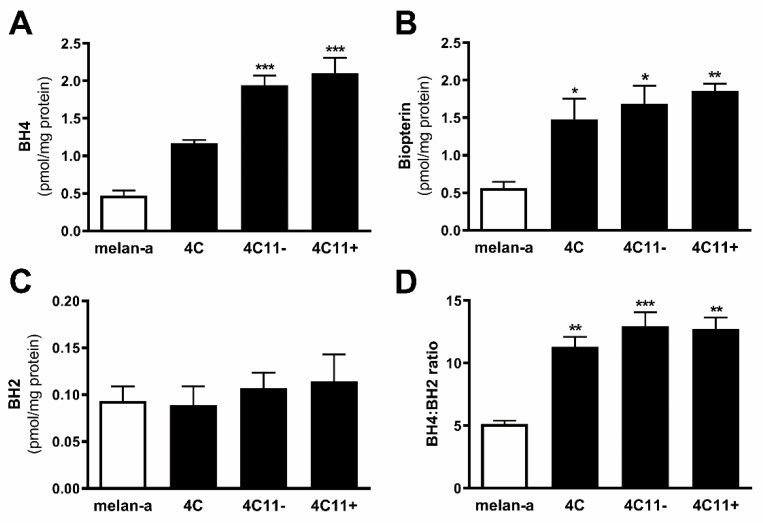
Tetrahydrobiopterin metabolism is altered along with melanoma progression. (**A**) The concentration of BH4, (**B**) BH2, (**C**) BH4:BH2 ratio, and (**D**) total biopterin in melan-a melanocytes (Ma), 4C pre-malignant melanocytes, 4C11− non-metastatic melanoma cells, and 4C11+ metastatic melanoma cells were evaluated by HPLC. Values are reported in the bar graphs and expressed as the means ± S.D. The experiments were performed in triplicate and *p* values were based on the one-way ANOVA test followed by Bonferroni post-test; *, *p* < 0.05; **, *p* < 0.01; ***, *p* < 0.001.

**Figure 3 ijms-22-09556-f003:**
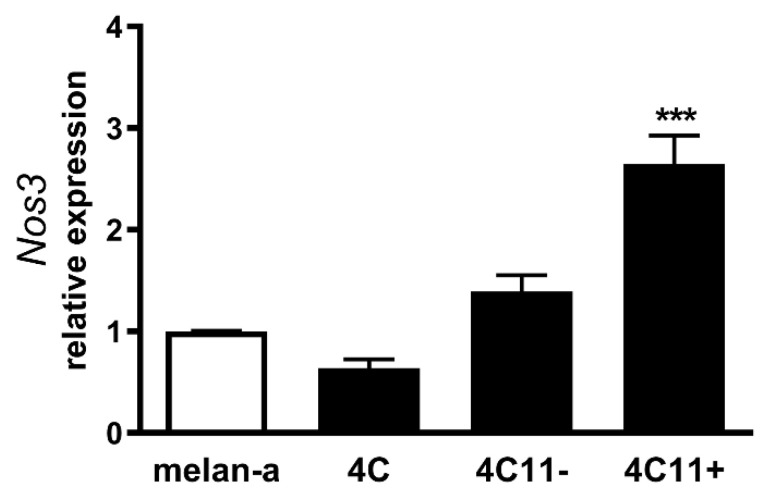
Increased expression of endothelial nitric oxide synthase in metastatic melanoma cells. Relative endothelial nitric oxide synthase mRNA expression was analyzed in melan-a melanocytes (Ma), 4C pre-malignant melanocytes, 4C11− non-metastatic melanoma cells, and 4C11+ metastatic melanoma cells by real-time qPCR using specific primers. Values are reported in the bar graphs and expressed as the means ± S.D. The experiments were performed in triplicate and *p* values were based on the one-way ANOVA test followed by Bonferroni post-test; ***, *p* < 0.001.

**Figure 4 ijms-22-09556-f004:**
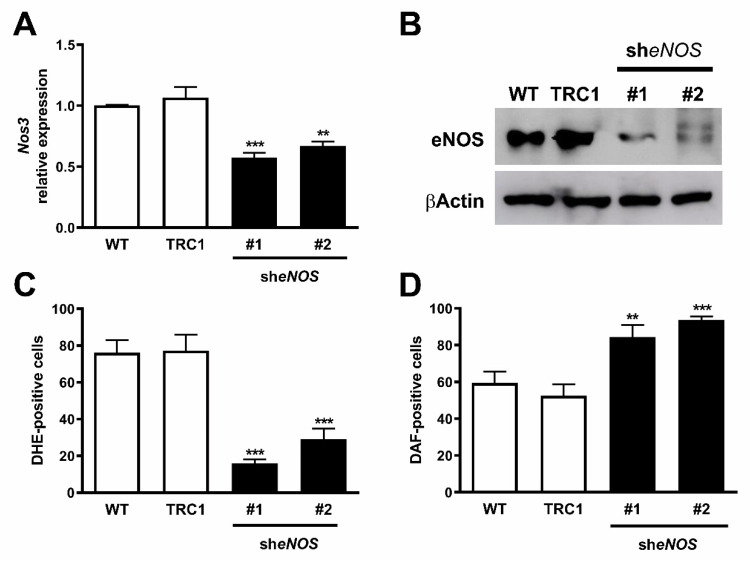
Endothelial nitric oxide synthase downregulation induced increased nitric oxide and reduced superoxide anion levels. Knockdown of eNOS was evaluated by (**A**) RT-qPCR and (**B**) Western blotting in wild-type 4C11+ metastatic melanoma cells (WT) and after transduction with viral particles containing two different shRNA sequences for eNOS (she*NOS* #1 and #2) or control non-target shRNA (TRC1). (**C**) Superoxide anion and (**D**) nitric oxide amount was analyzed using DHE or DAF, respectively, by flow cytometry. Values are reported in the bar graphs and expressed as the means ± S.D. The experiments were performed in triplicate and *p* values were based on the one-way ANOVA test followed by Bonferroni post-test; **, *p* < 0.01; ***, *p* < 0.001.

**Figure 5 ijms-22-09556-f005:**
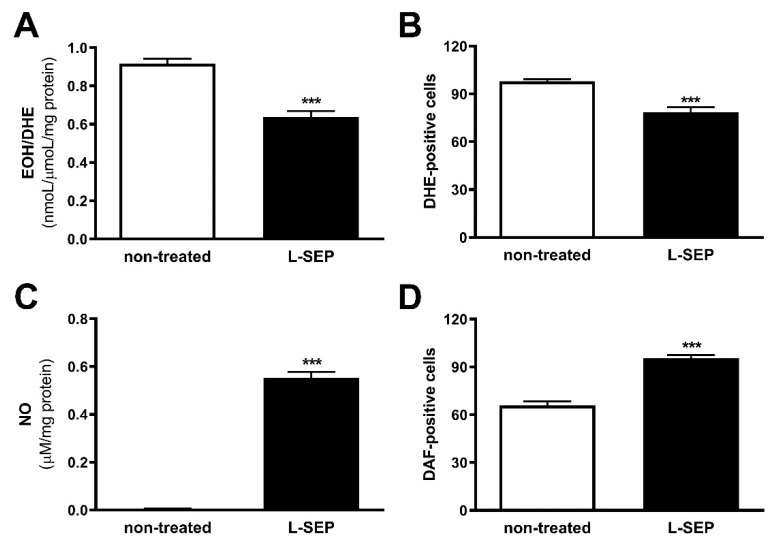
L-sepiapterin treatment restored endothelial nitric oxide synthase function. 4C11+ metastatic melanoma cells were treated or not (non-treated) with 40 μM L-sepiapterin (L-SEP) for two hours, and superoxide anion levels were analyzed using (**A**) DHE by HPLC or (**B**) flow cytometry, and nitric oxide amount were evaluated by (**C**) NO analyzer or by (**D**) flow cytometry using DAF. Values are reported in the bar graphs and expressed as the means ± S.D. The experiments were performed in triplicate and *p* values were based on the one-way ANOVA test followed by Bonferroni post-test; ***, *p* < 0.001.

**Figure 6 ijms-22-09556-f006:**
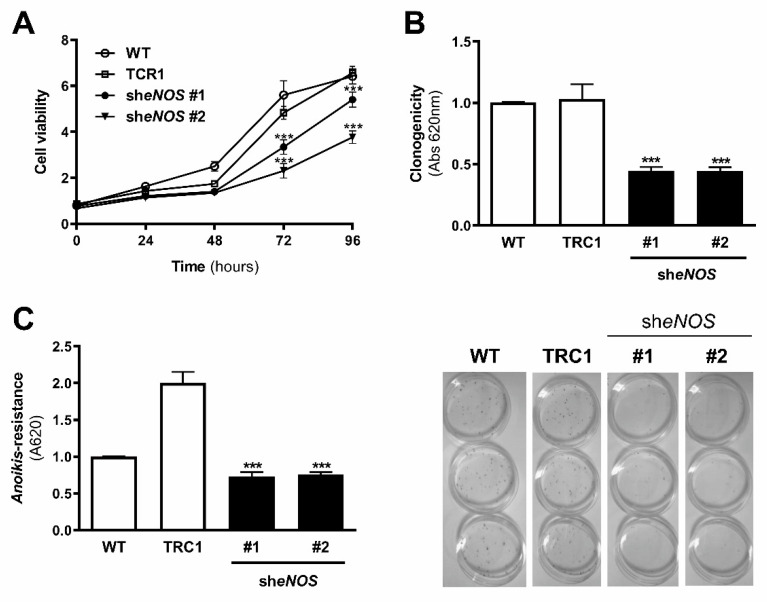
Endothelial nitric oxide synthase knockdown reduced melanoma cell growth in vitro. (**A**) Cell viability, (**B**) clonogenic capability, and (**C**) *anoikis* resistance were analyzed in 4C11+ metastatic melanoma cells WT, TRC1, she*NOS*#1, and she*NO*S#2. Cell viability was evaluated for 24, 48, 72, and 96 h by MTT. The clonogenic capability was determined by estimating clone formation after 9 days of cell culture and *anoikis* resistance by maintaining melanoma cells in suspension for 96 h and evaluating viable cells by MTT. Values are reported in the bar graphs and expressed as the means ± S.D. The experiments were performed in triplicate and *p* values were based on the one-way ANOVA test followed by Bonferroni post-test or on the two-way ANOVA test followed by Bonferroni post-test; ***, *p* < 0.001.

**Figure 7 ijms-22-09556-f007:**
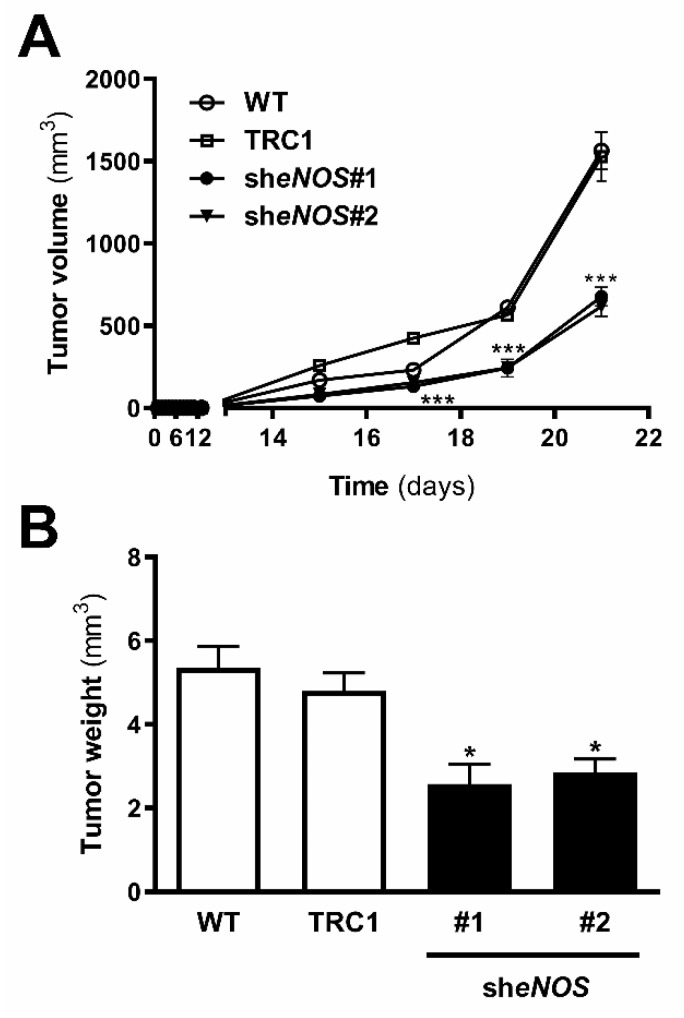
In vivo melanoma growth was impaired by endothelial nitric oxide synthase downregulation. 4C11+ metastatic melanoma cells were subcutaneously inoculated into C57BL/6 mice and (**A**) tumor weight and (**B**) volume were determined. WT: wild type 4C11+ metastatic melanoma cell line; TRC1: control non-target shRNA; she*NOS*#1 and whe*NOS*#2: clones silenced for eNOS. Values are reported in the bar graphs and expressed as the means ± S.D. The experiments were performed in triplicate and *p* values were based on the one-way ANOVA test followed by the Bonferroni post-test or two-way ANOVA test followed by Bonferroni post-test; *, *p* < 0.05, ***, *p* < 0.001.

**Figure 8 ijms-22-09556-f008:**
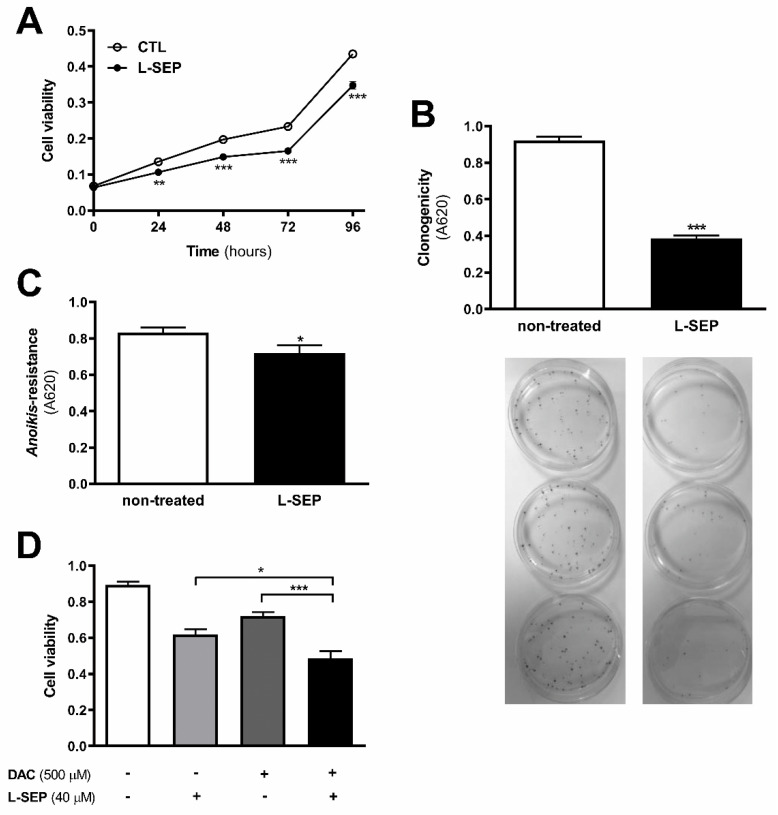
L-sepiapterin impaired cell growth and improved sensitivity to apoptosis of metastatic melanoma cells. 4C11+ metastatic melanoma cells were treated (+) or not (non-treated) (-) with 40 μM L-sepiapterin (L-SEP) and (**A**) cell viability, (**B**) clonogenic capability, (**C**) *anoikis* resistance, and (**D**) dacarbazine (DAC) treatment sensitivity were evaluated. WT: wild type 4C11+ metastatic melanoma cell line; TRC1: control non-target shRNA; she*NOS*#1 and she*NOS*#2: clones silenced for eNOS. Values are reported in the bar graphs and expressed as the means ± S.D. The experiments were performed in triplicate and *p* values were based on the one-way ANOVA test followed by the Bonferroni post-test or on the two-way ANOVA test followed by Bonferroni post-test; *, *p* < 0.05; **, *p* < 0.01; ***, *p* < 0.001.

**Figure 9 ijms-22-09556-f009:**
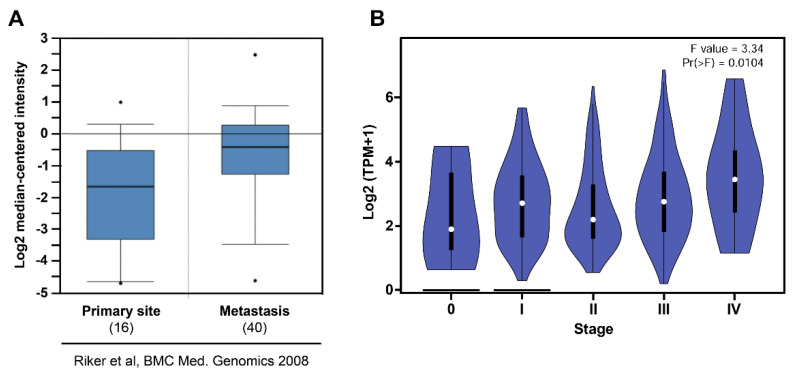
Increased endothelial nitric oxide synthase expression is correlated with human melanoma aggressiveness. (**A**) Melanoma metastasis (n = 40) expresses higher levels of *NOS3* compared to the primary site (n = 16). Microarray data by the Riker study was obtained from the Oncomine database as indicated (GSE7553). Boxes represent the interquartile range (25th–75th percentile). The bars denote the median; ^•^
*p* < −1 × 10^4^. (**B**) Increased *NOS3* expression along with melanoma progression (n = 48). Data regarding melanoma progression was obtained from gepia2.

## Data Availability

Not applicable.

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
