# Peer review of "Metastatic Melanoma Progression Is Associated with Endothelial Nitric Oxide Synthase Uncoupling Induced by Loss of eNOS:BH4 Stoichiometry"

_ijms, 2021, doi:10.3390/ijms22179556_

Round 1

Reviewer 1 Report

 The aim of this paper, entitled “Metastatic melanoma progression is associated with endothelial nitric oxide synthase uncoupling induced by loss of 3 eNOS:BH4 stoichiometry” is to investigate the relationship between the disfunction of endothelial nitric oxide synthase and melanoma progression.

The understanding of the mechanisms underlying the progression of melanoma is fundamental, given the high metastatic potential of this neoplasm. However, the work needs some changes before it is considered for publication

The main criticisms are as follows:

Melanoma mortality has decreased in recent years, due to the new therapeutic approaches such as immunotherapy and target therapy. The first part of the introduction should take into account this aspect.

The role of oxidation in the pathogenesis of other neoplasms favored by UV exposure, such as SCC, should be better discussed.

The paragraphs that, in the Results section, report data relating to previous works already published should be integrated within the Introduction section.

Authors should better specify characteristics of “pre-malignant 4C melanocytes”; are cells with characteristics similar to normal melanocytes, dysplastic nevi or malignant melanoma?

It could be interesting to verify the expression of eNOS on melanomas at different disease stages. Can the authors produce data in this regard, even from the literature?

Author Response

The responses are attached. 

Reviewer 2 Report

The manuscript titled “Metastatic melanoma progression is associated with endothelial nitric oxide synthase uncoupling induced by loss of eNOS:BH4 stoichiometry” by de Melo et al described endothelial nitric oxide synthase(eNOS) uncoupling in metastatic melanoma cells expressing the genes of de novo biopterin synthesis pathway Gch1, Pts and Spr, and high BH4 concentration and BH4:BH2 ratio. Increased expression of eNOS, altering the stoichiometry balance between eNOS and BH4, contributing to NOS uncoupling has been reported. Experiments have been performed in vitro and in vivo.

The manuscript is very professionally written and the experiments are also conducted professionally. Nevertheless, some aspects of the work should be improved.

In Figure 1 Relative mRNA levels of the enzymes from de novo tetrahydrobiopterin via, Gch1, Pts, Spr and Gfrp, were determined in melan-a melanocytes (Ma), 4C pre-malignant melanocytes, 4C11- non-metastatic melanoma cells and 4C11+ metastatic melanoma cells by real time qPCR using specific primers. The experiments were performed in triplicate. Nevertheless, the experiments seem to be performed in just a single cell line (one melan-a melanocytes (Ma) cell line, one  4C pre-malignant melanocyte cell line etc….). Different cell lines should be used as biological replicate, or a different model should be used to validate these results.  The same limitation as regarding the biological replicate can be observed also in Figure 2 and 3

Figure 5. the error bar is very small, please show the raw data to demonstrate the reproducibility of the data

Figure 8. Here just one cell line is used

Figure 9 only one dataset is shown. it would be interesting to have another dataset for example from TCGA or a validation on real patient samples

Author Response

The manuscript revision is attached. 

Round 2

Reviewer 2 Report

requests were met